# Metabolomic profiling of oxalate-degrading probiotic *Lactobacillus acidophilus* and *Lactobacillus gasseri*

**Casey A. Chamberlain, Marguerite Hatch, Timothy J. Garrett** *

Department of Pathology, Immunology and Laboratory Medicine, University of Florida, Gainesville, FL, United States of America

* tgarrett@ufl.edu

**Data Availability Statement:** Data from this study is available at www.metabolomicsworkbench.org (data track ID 1809).

**Funding:** TG received 2R01DK088892-05A1 from the National Institutes of Health, www.nih.gov. The

## Abstract

Oxalate, a ubiquitous compound in many plant-based foods, is absorbed through the intestine and precipitates with calcium in the kidneys to form stones. Over 80% of diagnosed kidney stones are found to be calcium oxalate. People who form these stones often experience a high rate of recurrence and treatment options remain limited despite decades of dedicated research. Recently, the intestinal microbiome has become a new focus for novel therapies. Studies have shown that select species of *Lactobacillus*, the most commonly included genus in modern probiotic supplements, can degrade oxalate *in vitro* and even decrease urinary oxalate in animal models of Primary Hyperoxaluria. Although the purported health benefits of *Lactobacillus* probiotics vary significantly between species, there is supporting evidence for their potential use as probiotics for oxalate diseases. Defining the unique metabolic properties of *Lactobacillus* is essential to define how these bacteria interact with the host intestine and influence overall health. We addressed this need by characterizing and comparing the metabolome and lipidome of the oxalate-degrading *Lactobacillus acidophilus* and *Lactobacillus gasseri* using ultra-high-performance liquid chromatography-high resolution mass spectrometry. We report many species-specific differences in the metabolic profiles of these *Lactobacillus* species and discuss potential probiotic relevance and function resulting from their differential expression. Also described is our validation of the oxalate-degrading ability of *Lactobacillus acidophilus* and *Lactobacillus gasseri*, even in the presence of other preferred carbon sources, measuring *in vitro* $^{14}$C-oxalate consumption via liquid scintillation counting.

## Introduction

Probiotics, defined by the World Health Organization (WHO) as "live microorganisms which when administered in adequate amounts, confer a health benefit on the host" [1, 2], have become widespread in the global health market. Often sold as foods or dietary supplements [3], many different probiotics exist that contain various cocktails of bacteria formulated to deliver specific health benefits ranging from immune system support [4], gastrointestinal regularity [5], serum cholesterol control [6], management of allergic diseases [7], and even relief of mental ailments

funders had no role in study design, data collection and analysis, decision to publish, or preparation of the manuscript.

**Competing interests:** The authors have declared that no competing interests exist.

**Abbreviations:** BMP, Bis(monoacylglycero) Phosphate; CER, Ceramide; CL, Cardiolipin; DG, Diacylglyercol; DGDG, Digalactosyldiacylglycerol; HBMP, Hemibismonoacylglycerophosphate; HexCer-NS, Hexosylceramide Non-Hydroxy Fatty Acid-Sphingosine; LPC, Lyosphosphatidylcholine; LSM, Lysosphinomyelin; MGDG, Monogalactosyldiacylglycerol; OxPG, Oxidized Phosphatidylglycerol; PA, Phosphatidic Acid; PAzePC, Oxidized Phosphatidylethanolamine 1-palmitoyl-2-azelaoyl-sn-glycero-3-phosphocholine; PC, Phosphatidylcholine; PE, Phosphatidylethanolamine; PG, Phosphatidylglycerol; PI, Phosphatidylinositol; PS, Phosphatidylserine; SO, Sphingosine; TG, Triacylglycerol.

such as anxiety and depression [8]. *Lactobacillus* is the most commonly-included genus of bacteria among typical probiotics [9]. Among the >200 species of these Gram-positive, rod-shaped microorganisms known to exist [10], over 50 have been shown to sustainably colonize the intestines of healthy individuals [11], although they are believed to be a minority among the projected 40,000 species that comprise the intestinal microbiome [12, 13]. Proposed health benefits linked to *Lactobacillus* vary significantly by species and strain [14]. Recently, interest has turned to *Lactobacillus* regarding its ability to degrade oxalate in the intestine [15, 16].

Oxalate is a toxic compound introduced to humans exogenously through the diet and endogenously through natural metabolism in the liver [17, 18]. Dietary oxalate, not metabolized by humans [19], is absorbed across the intestinal epithelium and precipitates with calcium in excreted urine to form calcium oxalate kidney stones [20]. Among all urinary stones, which cost the economy over $10 billion annually to treat [21], approximately 80% are calcium oxalate [20]. Intestinal absorption of oxalate contributes significantly to urinary oxalate levels [17], a primary risk factor for nephrolithiasis [16, 20]. Consequently, increased attention has turned to intestinal bacteria that are able to degrade dietary oxalate for their potential as future probiotic therapies for urinary stone formation and other oxalate conditions, such as the rare genetic disease Primary Hyperoxaluria (PH) [22–25]. In this report, we emphasize *Lactobacillus acidophilus* (*L. acidophilus*) and *Lactobacillus gasseri* (*L. gasseri*), two well-studied, common probiotic species with unique associations to human health [26]. *L. acidophilus* is perhaps the most well-recognized probiotic species and has been associated with many health benefits, including lowering total and LDL cholesterol [27, 28], reduction of symptoms of gastrointestinal ailments such as diarrhea [29, 30] and irritable bowel syndrome [31, 32], prevention of vaginal infections [33], alleviation of allergy symptoms [34], immune response regulation [35], and others. *L. gasseri*, although not as popular as *L. acidophilus*, has also been linked to many of these same health effects and is commonly studied for its purported association with weight loss [36, 37]. Both *L. acidophilus* and *L. gasseri* have shown extraordinary potential as future probiotics for oxalate diseases. A previous report examining oxalate degradation by *Lactobacillus* indicates that out of 60 strains tested from 12 different species, strains of *L. acidophilus and L. gasseri* were most efficient in their ability to degrade oxalate *in vitro* [15]. Furthermore, every strain tested from both these species showed the capability to degrade oxalate, whereas other species have some strains which did not show degradation [15]. Perhaps the most significant evidence supporting the probiotic potential of *L. acidophilus* and *L. gasseri* is from a study by *Hatch et al* showing their ability to reduce 24-hr urinary oxalate excretion in a mouse model of PH by 34% and 32%, respectively, as a result of intestinal colonization [16]. This serves as compelling evidence that these microbes should be further investigated as potential probiotic remedies for diseases of oxalate. Elucidating the unique metabolic properties of *L. acidophilus* and *L. gasseri* is essential to complete our understanding of the role these species play as symbiotic inhabitants of the human intestine as well as the cumulative health effect potentially delivered to the host as a probiotic. This investigation serves to define the species-specific biochemical qualities of *L. acidophilus* and *L. gasseri* by characterizing and comparing their metabolomic and lipidomic profiles using ultra-high-performance liquid chromatography-high resolution mass spectrometry (UHPLC-HRMS). For this study, we analyzed the same *L. acidophilus* and *L. gasseri* isolates demonstrated by Hatch *et al* to significantly reduce PH-model urinary oxalate *in vivo*. We discuss both commonalities and significant differences in the expression of many compounds between *L. acidophilus* and *L. gasseri* as well as the potential biological and probiotic relevance of significant features. Additionally, we report our investigation and confirmation of the oxalate-degrading ability of *L. acidophilus* and *L. gasseri* in the presence of other preferred carbon sources measuring *in vitro* [14]C-oxalate consumption via liquid scintillation counting.

## Methods

### Cell Culture, harvest, and Lysis

Pure cultures of *L. acidophilus* (ATCC™ 4357) and *L. gasseri* (ATCC™ 33323), the same isolates used by Hatch *et al* [16] obtained from the American Type Culture Collection (ATCC™), were grown anaerobically from frozen 10% glycerol stocks at 37°C in deMan, Rogosa and Sharpe (MRS) medium [15] supplemented with 20 mM oxalate and 1% glucose. Using a sterile syringe and needle, 8 anaerobic bottles containing 75 mL medium were each inoculated with 150 μL glycerol stock for each species. Cultures were briefly shaken and allowed to incubate overnight at 37°C for 24 hours. After incubation, cultures were harvested as individual biological replicates (n = 8 per species) using a process similar to our previously reported harvest and lysis method [38, 39], which we describe here. Cultures were removed from the 37°C incubator and centrifuged at 15,180×g, 4°C for 5 min to isolate bacterial pellets by discarding the conditioned medium supernatants. Pellets were washed 3 times by repeated resuspension in 6 mL 100 mM $KH_2PO_4$-based lysis buffer [40] followed by centrifugation. After the third wash, pellets were dried, weighed, resuspended in lysis buffer to a normalized concentration of 75 mg/mL, and transferred to 15 mL polypropylene (PP) vials. Cells were lysed by sonication while chilled in an ice bath using a Sonic Dismembrator Model 500 with a Branson Sonicator Probe (Thermo Fisher Scientific, Waltham, MA, USA) by the following method: 30% amplitude for 30 sec, 1 min cool-down, 60% amplitude for 30 sec, 2 min cool-down, 60% amplitude for 15 sec. Cell lysates were immediately frozen at −80°C to ensure their stability and were briefly held frozen (approximately 1 month) until needed for extraction, all samples being stored for an equal period of time. In our experience, lysates of this nature suspended in $KH_2PO_4$-based lysis buffer are stable for metabolomics analyses for several years.

### Metabolite extraction

Wherever possible, during the metabolite and lipid extractions, samples were chilled on ice and protected from light. All reagents were LC-MS grade (Thermo Fisher Scientific). Metabolites were extracted by protein precipitation similarly to our previous work [38] using the following procedure. From each normalized cell lysate sample, 100 μL was transferred to a 1.6 mL PP vial. Extraction blanks were also included for downstream data filtering using 100 μL lysis buffer and were treated identically to biological samples. To each sample, 20 μL of internal standard mixture (Acros Organics, Fairlawn, NJ, USA) in 0.1% formic acid in water was added followed by brief vortexing–Creatine(1-methyl-$D_3$), D-Leucine-$D_{10}$, L-Tryptophan-2,3,3-$D_3$, L-Tyrosine-$^{13}C_6$, L-Leucine-$^{13}C_6$, L-Phenylalanine-$^{13}C_6$, N-BOC-L-tert-Leucine, N-BOC-L-Aspartic Acid, Succinic Acid-2,3,3,3-$D_4$, Salicylic Acid-$D_6$, Caffeine-(1-methyl-D3) (each 4 μg/mL), Propionic Acid-$^{13}C_3$ (8 μg/mL), L-Tryptophan-2,3,3-$D_3$ (40 μg/mL). Next, 800 μL 8:1:1 acetonitrile:methanol:acetone was added to precipitate protein. Samples were again briefly vortexed and incubated on ice for 30 min. Protein content was pelleted by centrifugation at 20000×g, 4°C for 10 min, and 750 μL supernatants were transferred to new 1.6 mL PP vials. Supernatants were dried under nitrogen at 30°C and resuspended in 100 μL 0.1% FA in water. Samples were centrifuged at 20000×g, 4°C for 10 min to pellet any remaining protein, and 50 μL supernatants were transferred to glass LC vials for UHPLC-HRMS analysis.

### Lipid extraction

Lipids were extracted using a modified version of the Folch method [41] similarly to our previous work [38] using the following process which we describe in detail here. From each normalized cell lysate sample, 150 μL was transferred to a 12 mL glass vial. Extraction blanks were included for downstream data filtering using 150 μL lysis buffer and were treated identically to biological

samples. To each sample, 20 μL of internal standard mix (Avanti Polar Lipids, Alabaster, AL, USA) was added followed by brief vortexing–LPC(17:0), PC(17:0/17:0), PG(14:0/14:0), PE(15:0/15:0), PS(14:0/14:0), TG(15:0/15:0/15:0), PI(8:0), SM(d18:1/17:0), CER(d18:1/17:0), DG(14:0/14:0), CL(15:0(3)-16:1), SO(d17:1), PAzePC, CER(Glycosyl($\beta$) C12), BMP(14:0 (*S*,*R*)), LSM (d17:1), 5 μg/mL each in 2:1 chloroform:methanol. Next, 400 μL methanol was added to each sample followed by vortexing, then 800 μL chloroform was added. Samples were vortexed and incubated on ice for 20 min, with vortexing at 10 and 20 min, followed by addition of 200 μL water. Samples were briefly vortexed and incubated on ice for 10 min with vortexing at 5 and 10 min. Separation of the organic and aqueous layers was achieved by centrifugation at 3260×*g*, 4°C for 10 min. The organic (bottom) layer containing lipid content was transferred to a new 12 mL glass vial in two steps: first with removal of 800 μL of the original organic layer, followed by another removal of 400 μL after re-extracting the remaining aqueous layer with 400 μL 2:1 chloroform:methanol by incubating on ice for 10 min and centrifugation at 3260×*g*, 4°C for 10 min. Lipid extracts were dried under nitrogen at 30°C and reconstituted in 300 μL isopropanol. Samples were centrifuged at 3260×*g*, 4°C for 10 min to pellet any residual protein, and 250 μL supernatants were transferred to glass LC vials for UHPLC-HRMS analysis.

## Analytical instrumentation and methodology

Metabolomics and lipidomics analyses by UHPLC-HRMS were performed on a Thermo Q Exactive Orbitrap Mass Spectrometer with heated electrospray ionization source coupled to a Dionex Ultimate 3000 UHPLC system (Thermo Scientific). Parameters for the metabolomics analysis will be discussed first, followed by the lipidomics analysis. For the metabolomics analysis, reverse phase chromatography with gradient elution was employed using an ACE Excel 2 C18-PFP column (100mm × 2.1mm, 2.0μm) (Advanced Chromatography Technologies, Ltd, Scotland). Gradient elution was performed with 0.1% formic acid in water as solvent A and acetonitrile as solvent B, at a flow rate of 0.35 mL/min, as such: 0–3 min: 100% A, 3–13 min: 100% → 20% A, 13–16.5 min: 20% A, 16.5–20 min: 100% A at 0.6 mL/min (column flush & equilibration). Injection volume was 5 $\mu$L. Data were acquired in both positive and negative ion mode by full scan analysis from *m/z* 70–1000 at 35000 mass resolution. For the lipidomics analysis, reverse phase chromatography was again employed using an AQUITY UPLC BEH C18 column (50mm × 2.1mm, 1.7μm) (Waters Corporation, Milford, MA, USA) preceded by a corresponding VanGuard pre-column (Waters Corporation). Gradient elution was performed with 60:40 acetonitrile:water with 0.1% formic acid and 10mM ammonium formate as solvent A and 90:8:2 isopropanol:acetonitrile:water with 0.1% formic acid and 10mM ammonium formate as solvent B, at a flow rate of 0.5 mL/min, as such: 0–1 min: 80% A, 1–3 min: 80% → 70% A, 3–4 min: 70% → 55% A, 4–6 min: 55% → 40% A, 6–8 min: 40% → 35% A, 8–10 min: 35% A, 10–15 min: 35% → 10% A, 15–17 min: 10% → 2% A, 17–18 min: 2% A, 18–19 min: 2% → 80% A, 19–23 min: 80% A (column flush & equilibration). Injection volume was 10 $\mu$L. Data were acquired in both positive and negative ion mode by full scan (70000 mass resolution), data-dependent MS/MS (35000 mass resolution), and all-ion fragmentation MS/MS (70000 mass resolution) analyses from *m/z* 200–2200. Using pooled samples for each species, we employed iterative exclusion analysis, consisting of repeated data-dependent MS/MS analysis with successive exclusion of detected features, to allow detection of lower abundance lipid species in both positive (6 rounds) and negative (4 rounds) ion mode [42].

## Data processing

For quality control purposes, the performance of spiked internal standards was assessed in all samples. Excellent reproducibility was verified with standards showing relative standard

deviations < 10%. Parameters for the metabolomics data processing will be discussed first, followed by the lipidomics processing. Metabolomics data files were converted from .raw to .mzxml format using RawConverter [43]. MZmine 2 was employed for all processing involved in peak picking, chromatographic alignment, and metabolite identification [44]. Metabolites were identified by *m/z* (5 ppm) and elution time (±0.2 min) matching to our method-specific internal library produced from pure analytical standards. Non-detected signals were replaced with half the minimum signal intensity value in the dataset [45]. Data were filtered to remove features with ≥10% signal contributed from the background as determined by comparison to the extraction blanks [46]. Signal intensities were median-normalized [47] and autoscaled [48]. For the lipidomics data, all processing, including file format conversion, peak picking, chromatographic alignment, and lipid identification (MS/MS fragmentation spectral matching to *in silico* databases) was performed using LipidMatch software [46]. Missing value replacement, as well as data filtration, normalization, and autoscaling, were performed similarly to the metabolomics data.

## Statistical analysis

MetaboAnalyst 4.0 was used for statistical analysis and figure generation [49]. All *p*-values were determined using the two-tailed, unpaired Student's t-test assuming equal variance on the normalized, scaled dataset and adjusted for the false discovery rate using the Bonferroni-Holm correction [50]. In this report, we define significance with a *p*-value threshold of ≤0.001.

## Determination of oxalate degradation by liquid scintillation counting

Two 100 mL anaerobically sealed vials with 75 mL sterile MRS medium [15], supplemented with 20 mM oxalate and 1% glucose, were each spiked with 3 μCi of $^{14}$C-oxalate (Vitrax, Placentia. CA, USA). Following a thorough mixing of the vial contents, two 100 μL aliquots were removed from each 100 mL vial for liquid scintillation counting prior to inoculating with either *L. acidophilus* or *L. gasseri*. These vials were incubated at 37°C for 3 days, after which time duplicate 100 μL aliquots were removed from each culture and placed in liquid scintillation vials followed by acidification of the aliquot with HCl to induce volatilization of $^{14}$CO$_2$ in the fume hood for a 24-hr period. Scintillation fluid was then added to the vials followed by liquid scintillation counting using a LS 6500 Multi-Purpose Scintillation Counter (Beckman Coulter, Brea, CA, USA). Final counts were compared to those pre-inoculation to quantify the remaining counts and calculate the percent-degradation of the original oxalate substrate by *L. acidophilus* and *L. gasseri*.

## Results and discussion

### Metabolomics analysis

The refined metabolomics dataset consisted of a total of 2077 features detected between positive and negative ion mode. Significant differences between the metabolomes of these bacteria were observed at the global scale as well as the level of individual analytes. S1A Fig demonstrates clear separation of the general metabolomes of *L. acidophilus* and *L. gasseri* by Principal Component Analysis (PCA), mainly accounted for by PC1, with 76.8% of the variance described in 2 PCs. Fig 1 portrays the high metabolomic diversity between *L. acidophilus* and *L. gasseri* with a volcano plot depicting the distribution of detected features by the magnitude and significance of their differential signal intensities. Among all detected features, 1508 (72.6%) showed a statistically significant difference in their relative intensities between species with 702 (46.6%) showing elevated expression in *L. acidophilus* and 806 (53.4%) showing

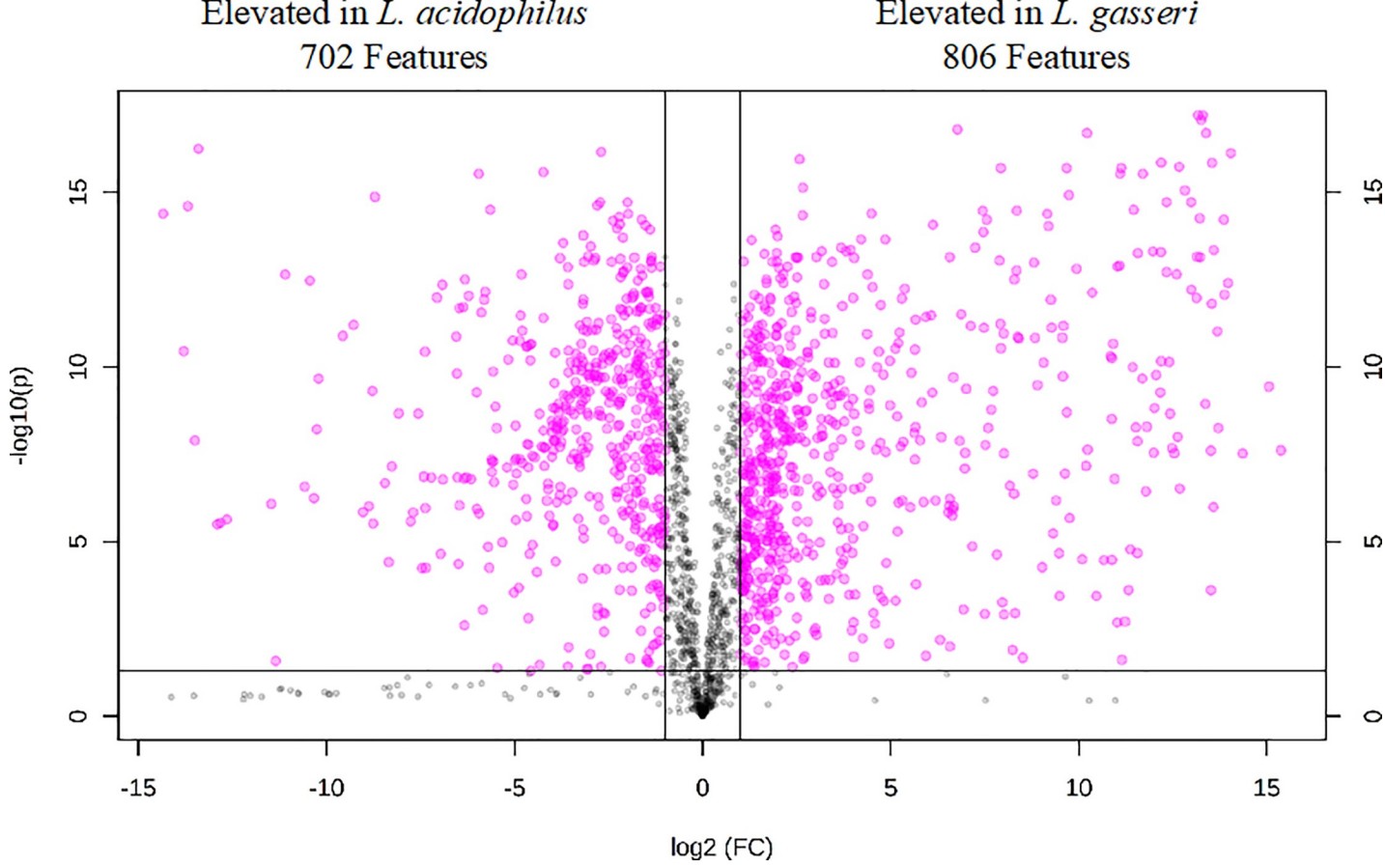

**Fig 1. Volcano plot depicting high metabolomic diversity between *L. acidophilus* and *L. gasseri*.** A total of 1508 features (72.6%) showing a statistically significant difference in their relative intensities between species with 702 (46.6%) showing elevated expression in *L. acidophilus* and 806 (53.4%) showing elevated expression in *L. gasseri*.

elevated expression in *L. gasseri*. A total of 97 metabolites were identified by *m/z* and retention time matching to our internal library as listed in S1 Table with the magnitude and significance of the difference in their relative intensities between species. Table 1 details the top-25 significant identified metabolites. Among the most significant metabolites were several components of the Krebs Cycle, including succinic acid ($p = 1.85 \times 10^{-13}$, 4.3-fold higher intensity in *L. acidophilus*), fumaric acid ($p = 6.25 \times 10^{-12}$, 623.2-fold higher intensity in *L. acidophilus*), citric acid ($p = 6.70 \times 10^{-12}$, 139.9-fold higher intensity in *L. gasseri*), and malic acid ($p = 6.74 \times 10^{-10}$, 3.3-fold higher intensity in *L. gasseri*). It is worth mentioning that although these bacteria were grown under anaerobic conditions, leading us to expect the Krebs Cycle was not functioning, our results indicate a more complex metabolic process that is different between these two anaerobes. Such differential expression of these key metabolic compounds could have a variety of biological implications. Fumaric acid showed the greatest difference between species by magnitude. *Lactobacillus* has been noted in the literature due to its potential for industrial mass production of fumaric acid by fermentation [51]. In addition, fumaric acid is known to have an antimicrobial effect, likely due to acidifying the extracellular pH and making the environment inhospitable to competing microorganisms [52–54]. High production of fumaric acid for this purpose could be one reason behind the association between intestinal colonization by *Lactobacillus* and protective host immunity from pathogenic gut bacteria. Additionally,

**Table 1. Top-25 identified metabolites of greatest significant difference in signal intensity between *L. acidophilus* and *L. gasseri*.**

| Metabolite Species | Exp Mass | Ion | Fold-Difference | p-value | Elevated Expression |
|---|---|---|---|---|---|
| Adenosine-5'-Diphosphate | 426.0230 | [M-H]- | 3.9 | 4.10E-15 | *L. acidophilus* |
| N-Acetylputrescine | 131.1179 | [M+H]+ | 7.3 | 7.49E-14 | *L. acidophilus* |
| Cytidine-5'-Diphosphate-glycerol | 476.0485 | [M-H]- | 12.0 | 1.38E-13 | *L. acidophilus* |
| Succinic Acid | 117.0197 | [M-H]- | 4.3 | 1.85E-13 | *L. acidophilus* |
| Gluconic Acid | 195.0511 | [M-H]- | 56.2 | 1.17E-12 | *L. acidophilus* |
| Betaine | 118.0863 | [M+H]+ | 2.0 | 3.16E-12 | *L. acidophilus* |
| Fumaric Acid | 115.0038 | [M-H]- | 623.2 | 6.25E-12 | *L. acidophilus* |
| Asparagine | 133.0607 | [M+H]+ | 8.4 | 9.41E-12 | *L. acidophilus* |
| Choline | 104.1070 | [M+H]+ | 1.7 | 2.16E-11 | *L. acidophilus* |
| 5-Hydroxymethyl-2-furaldehyde | 127.0391 | [M+H]+ | 6.1 | 4.39E-11 | *L. acidophilus* |
| Trimethyllysine | 189.1597 | [M+H]+ | 1.8 | 1.02E-10 | *L. acidophilus* |
| Glucuronic Acid | 193.0356 | [M-H]- | 11.9 | 1.41E-10 | *L. acidophilus* |
| Glycerophosphocholine | 258.1098 | [M+H]+ | 2.6 | 1.65E-10 | *L. acidophilus* |
| Aspartic Acid | 134.0447 | [M+H]+ | 3.9 | 1.76E-10 | *L. acidophilus* |
| Methionine | 150.0584 | [M+H]+ | 4.3 | 2.99E-10 | *L. acidophilus* |
| Lysine | 147.1127 | [M+H]+ | 3.7 | 4.97E-10 | *L. acidophilus* |
| 3-Methyloxindole | 148.0757 | [M+H]+ | 2.5 | 8.53E-10 | *L. acidophilus* |
| Carnitine | 162.1125 | [M+H]+ | 1.5 | 2.14E-09 | *L. acidophilus* |
| Valine | 116.0717 | [M-H]- | 2.3 | 2.46E-09 | *L. acidophilus* |
| Citric Acid | 191.0200 | [M-H]- | 139.9 | 6.70E-12 | *L. gasseri* |
| N-Acetylglutamic Acid | 190.0709 | [M+H]+ | 3.9 | 7.21E-12 | *L. gasseri* |
| 3-Hydroxy-3-Methylglutaric Acid | 185.0421 | [M+Na]+ | 2.7 | 2.91E-11 | *L. gasseri* |
| Proline | 116.0706 | [M+H]+ | 4.3 | 6.00E-11 | *L. gasseri* |
| Malic Acid | 133.0145 | [M-H]- | 3.3 | 6.74E-10 | *L. gasseri* |
| Nicotinic Acid | 124.0395 | [M+H]+ | 10.7 | 1.21E-09 | *L. gasseri* |

differential expression of fumaric acid between *Lactobacillus* species as seen in this experiment could result in dissimilar protective capacity between such species. Fumaric acid has also been suggested to serve as a terminal electron acceptor in electron transport chains (ETC) of several species of anaerobic bacteria, including lactic acid bacteria, for enhanced growth in the absence of oxygen [55, 56]. The significantly different expression of fumaric acid between *L. acidophilus* and *L. gasseri* could suggest a differential reliance on this metabolic pathway, although to our knowledge, the utility of this proposed ETC in these species has not been reported or characterized in the literature. The high relative expression of citric acid by *L. gasseri*, serving as the second greatest difference by magnitude, is also interesting. Citric acid is a ubiquitous compound able to be produced by many different species of bacteria, some even extensively, under specific growth parameters [57]. Although the production of citric acid by *Lactobacillus* is not well-documented in the literature, several studies have shown that, under certain conditions, several species of *Lactobacillus* can use citric acid as an energy source and that the presence of citric acid in culture media has profound effects on their overall growth rate and metabolism [58–62]. The observed differential expression of citric acid between *L. acidophilus* and *L. gasseri* deserves further investigation. Gluconic acid ($p = 1.17 \times 10^{-12}$, 56.2-fold higher intensity in *L. acidophilus*) and glucuronic acid ($p = 1.41 \times 10^{-10}$, 11.9-fold higher intensity in *L. acidophilus*) also showed a significant difference between species. Gluconic acid and glucuronic acid are oxidation products of glucose, formed from oxidation at C1 and C6, respectively [63]. Both glucose derivatives have been shown to serve defensive functions in

bacteria. Gluconic acid is produced by *Pseudomonas* as a key antifungal metabolite [64], which would translate as a potentially important role for an intestinal *Lactobacillus* species in terms of providing host immunity. Glucuronic acid plays a detoxification role in humans by binding to hormones, drugs, and toxins, forming glucuronides to facilitate their transport and elimination from the body. This process, known as glucuronidation, involves glycosidic bond formation of glucuronic acid from uridine diphosphate-glucuronic acid with these compounds by UDP-glucuronosyltransferases and is an important method by which harmful substances are solubilized and cleared from the body [65, 66]. The production of glucuronic acid by *Lactobacillus* is indicative of a potential detoxification role these bacteria may play in the intestine. Since both gluconic acid and glucuronic acid were found to be elevated in *L. acidophilus*, it could be assumed that this species may be more effective in delivering their proposed health benefits to the human host as compared to *L. gasseri*, but this hypothesis requires further investigation to validate. Further work is needed to confirm the biological functionality of these discussed metabolites, as well as all other metabolites identified in this report, and characterize the nature of their differential expression between *L. acidophilus* and *L. gasseri*.

### Lipidomics analysis

As with the metabolomics analysis, significant differences between the lipidomes of *L. acidophilus* and *L. gasseri* were observed. S1B Fig demonstrates clear separation of the lipidomes of these bacteria by PCA, mainly accounted for by PC1, with 83.7% of the variance described in 2 PCs. A total of 71 lipid species were identified using a combination of MS/MS fragmentation and exact mass matching to *in silico* databases using LipidMatch open source software [46]. A complete list of identified lipids is presented in S2 Table along with the magnitude and significance of the difference in their relative intensities between species. Due to the nature of lipidomic analyses, a varying degree of overlap can sometimes present itself when making identifications because of the challenge of chromatographically resolving all the possible isomeric structures. Therefore, possible additional isomeric identifications to select lipids are provided. Among the 71 identified lipids, 59 showed a significant difference in their intensity between species with 27 showing elevated expression in *L. acidophilus* and 32 showing elevated expression in *L. gasseri*. The top-25 significant identified lipids are detailed in Table 2. We observed that diacylglycerols (DGs), digalactosyldiacylglycerol (DGDGs), and phosphatidylglycerols (PGs) showed the greatest representation among the significant lipids. Regarding DGs, the nature of their fatty acyl chains was observed to differ significantly between species. DGs elevated in *L. gasseri* all possessed only 16 and 18-carbon tails, except DG(8:0/18:1). However, among the DGs elevated in *L. acidophilus*, a greater diversity was seen with 17, 18, 19, and 20-carbon tailed species. Among the most significant are DG(18:1/20:1) ($p = 4.29{\times}10^{-12}$, 697.5-fold elevated intensity in *L. acidophilus*), DG(18:3/19:0) ($p = 4.29{\times}10^{-12}$, 15.1-fold elevated intensity in *L. acidophilus*), and DG(18:2/19:0) ($p = 9.39{\times}10^{-12}$, 38.8-fold elevated intensity in *L. acidophilus*). Furthermore, *L. acidophilus* appeared to exhibit a greater tendency to produce odd-chain lipids as 17 of the 19 odd-chain lipids detected were found to be elevated in *L. acidophilus*. Although the biological implications behind this observation are not immediately clear, it is known that Gram-positive bacteria can exhibit differential enzymatic biochemistry involved in lipid synthesis which can influencing the balance of even-versus-odd-chain fatty acid production [67]. Examining the general lipidomes of *L. acidophilus* and *L. gasseri*, we observed highly-differential class-level distribution of identified lipids as measured by summing the signal intensity of individual lipid classes (Fig 2). Both species displayed the same 5 lipid classes as the primary constituents (>98%) of their lipidomes: DGs, DGDGs, PGs, bis (monoacylglycero)phosphates (BMPs), and cardiolipins (CLs). All other detected lipid classes

**Table 2. Top-25 identified lipids of greatest significant difference in signal intensity between *L. acidophilus* and *L. gasseri*.**

| Lipid Species | Exp Mass | Ion | | Fold-Difference | p-value | Elevated Expression | Additional Isomeric Identifications |
|---|---|---|---|---|---|---|---|
| DG(18:1/20:1) | 666.6038 | [M+NH4]+ | 1 | 697.5 | 4.29E-12 | *L. acidophilus* | |
| DG(18:3/19:0) | 650.5719 | [M+NH4]+ | 1 | 15.1 | 4.29E-12 | *L. acidophilus* | DG(17:1/20:2) \| DG(17:2/20:1) |
| DG(18:2/19:0) | 652.5879 | [M+NH4]+ | 1 | 38.8 | 9.39E-12 | *L. acidophilus* | |
| DGDG(18:2/19:0) | 976.6933 | [M+NH4]+ | 1 | 53.6 | 4.84E-11 | *L. acidophilus* | DGDG(37:2) \| DGDG(17:2/20:0) \| DGDG(17:1/20:1) |
| DGDG(17:1/20:2) | 974.6768 | [M+NH4]+ | 1 | 78.8 | 7.28E-11 | *L. acidophilus* | DGDG(17:2/20:1) \| DGDG(18:3/19:0) \| DGDG(15:1/22:2) \| DGDG(37:3) |
| BMP(19:1/18:2) | 804.5748 | [M+NH4]+ | 1 | 4.9 | 1.36E-10 | *L. acidophilus* | BMP(18:1/19:2) \| PG(37:3) |
| DG(17:0/18:1) | 626.5724 | [M+NH4]+ | 1 | 5.9 | 1.36E-10 | *L. acidophilus* | |
| PG(18:1/20:1) | 820.6067 | [M+NH4]+ | 1,2 | 50.9 | 2.29E-10 | *L. acidophilus* | BMP(19:1/19:1) \| PG(38:2) \|\| BMP(18:1/20:1) |
| BMP(18:0/19:1) | 808.6065 | [M+NH4]+ | 1 | 7.1 | 6.59E-10 | *L. acidophilus* | PG(37:1) |
| HexCer-NS(d18:1/22:1) | 782.6506 | [M+H]+ | 2 | 1.7 | 6.59E-10 | *L. acidophilus* | |
| DGDG(17:0/18:1) | 950.6786 | [M+NH4]+ | 1 | 5.3 | 3.92E-09 | *L. acidophilus* | DGDG(35:1) |
| DGDG(18:1/19:0) | 978.7096 | [M+NH4]+ | 1 | 78.7 | 5.22E-09 | *L. acidophilus* | DGDG(37:1) |
| DGDG(18:1/18:1) | 989.6444 | [M+HCO2]- | 1 | 10.5 | 4.35E-12 | *L. gasseri* | |
| PG(18:1/18:1) | 773.5359 | [M-H]- | 1,2 | 6.4 | 1.45E-11 | *L. gasseri* | PG(18:0/18:2) |
| PG(18:1/18:2) | 771.5200 | [M-H]- | 1,2 | 6.0 | 1.63E-11 | *L. gasseri* | |
| PG(18:0/18:1) | 775.5512 | [M-H]- | 1,2 | 4.1 | 2.52E-11 | *L. gasseri* | |
| DGDG(18:1/18:2) | 987.6286 | [M+HCO2]- | 1 | 3.9 | 5.35E-11 | *L. gasseri* | |
| PG(18:1/18:3) | 769.5049 | [M-H]- | 1,2 | 6.6 | 5.84E-11 | *L. gasseri* | PG(18:2/18:2) |
| DGDG(16:0/18:1) | 963.6285 | [M+HCO2]- | 1 | 5.1 | 9.60E-11 | *L. gasseri* | |
| PG(8:0/18:1) | 635.3940 | [M-H]- | 1 | 6.9 | 6.26E-10 | *L. gasseri* | |
| CL(36:2)(36:2) | 1475.0696 | [M+NH4]+ | 1,2 | 30.0 | 1.13E-09 | *L. gasseri* | CL(36:1)(36:3) |
| PG(12:0/18:1) | 691.4572 | [M-H]- | 1 | 3.5 | 1.90E-09 | *L. gasseri* | |
| HexCer-NS(d18:1/18:1) | 726.5886 | [M+H]+ | 2 | 50.3 | 2.76E-09 | *L. gasseri* | |
| PG(16:1/18:1) | 745.5043 | [M-H]- | 1 | 6.7 | 5.28E-09 | *L. gasseri* | PG(16:0/18:2) |
| CL(18:1/18:1/18:1/18:1) | 727.5119 | [M-2H]2- | 1,2 | 4.2 | 7.17E-09 | *L. gasseri* | |

Detection: 1 = Data-Dependent (top 5) MS/MS fragment *m/z* match, 2 = All-Ion-Fragmentation MS/MS fragment *m/z* match

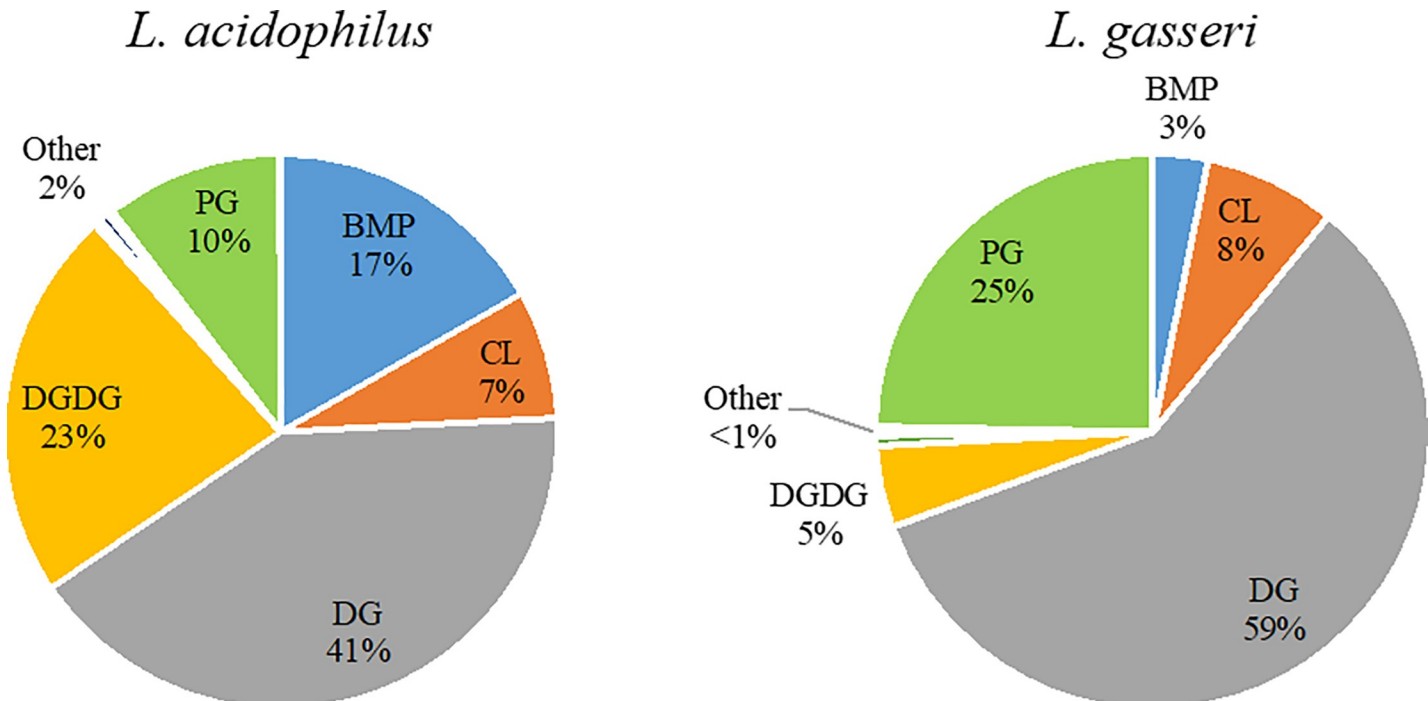

**Fig 2. Distribution of identified lipid species by class intensity sum for *L. acidophilus* and *L. gasseri*.** The majority of lipids were found to be diacylglycerols (DGs), digalactosyldiacylglycerols (DGDGs), phosphatidylglycerols (PGs), bis(monoacylglycero)phosphates (BMPs), and cardiolipins (CLs). Other detected lipid classes represented a minority (2% or less) of the total lipid signal. Significant variance in the relative ratios of these classes was observed between *L. acidophilus* and *L. gasseri*, with exception to CLs (7% and 8%, respectively), indicating the presence of species-specific lipid membrane compositions within *Lactobacillus*.

comprised a minority of the total lipid signal (≤2%) and are detailed in S2 Table. Although *L. acidophilus* and *L. gasseri* were found to share the same 5 core lipid classes, the ratios of these classes as a measure of the total lipid signal are significantly different between species. In both *L. acidophilus* and *L. gasseri*, DGs had the highest class representation at 41% and 59%, respectively. A difference of nearly 50% in the primary lipid class between two bacteria, especially within the same genus, is notable. In *L. acidophilus*, DGDGs represented 23% of the lipid signal, a sharp contrast to 5% in *L. gasseri*. PGs and BMPs were also very different between the species with PGs accounting for only 10% of the lipid signal in *L. acidophilus* and 25% in *L. gasseri*, and BMPs accounting for 17% of the lipid signal in *L. acidophilus* and 3% in *L. gasseri*. The only core lipid class that showed similar representation between species was CLs at 7% in *L. acidophilus* and 8% in *L. gasseri*. Gaining an understanding of the unique lipid profiles of these *Lactobacillus* species is important to understand their relationship with the host intestine. The composition of the cell structure directly affects how a bacterium responds to its environment, particularly through surface expression or secretion of key signaling and metabolic factors [68]. Certain *Lactobacillus* species have been shown to modify the composition of their lipid membranes as a protective mechanism against oxidative and thermal stress [69], salt exposure [70], and low pH [71]. Additionally, the nature of a bacterium's surface lipid content allows for modulation of immune responses, specifically activation of the innate immune response, due to interaction between microbe-associated molecular patterns (including lipids) on the microbe surface and pattern recognition receptors on the mucosal surface, triggering production of a variety of effector molecules [72]. Hence, the lipid profiles of candidate probiotic bacteria should be considered a significant point of interest as some species, as a result of their unique lipid makeup, may exhibit more robust colonization and delivery of probiotic

health benefits due to the extent at which they can adapt to environmental stressors and coexist within the host intestine. Further work is needed to confirm the biological functionality behind the differential expression observed in the *L. acidophilus* and *L. gasseri* lipidomes.

## Oxalate degradation by *L. acidophilus* and *L. gasseri*

Liquid scintillation counting evaluation of oxalate degradation by *L. acidophilus* and *L. gasseri* demonstrated significant degradation by both species. *L. acidophilus* showed 100% degradation of the $^{14}$C-oxalate with ~ 44% of counts remaining representing $^{14}$C-formate in the media from enzymatic $^{14}$C-oxalate degradation via oxalate decarboxylase [73]. *L. gasseri* showed ~ 72% of counts remaining, meaning it degraded ~50% of the $^{14}$C-oxalate in the media. Our findings are consistent with past experiments reporting oxalate degradation by both *L. acidophilus* and *L. gasseri* in the presence of other carbon sources with *L. acidophilus* being noted as a more efficient degrader [15, 16]. These results provide further evidence supporting the evaluation of these *Lactobacillus* species as potential probiotic remedies for oxalate pathologies.

## Conclusions

We conclude that the metabolomes and lipidomes of *L. acidophilus* and *L. gasseri* displayed appreciable differentiation both in terms of their general profiles and relative expression of individual compounds. Although we successfully identified 97 metabolites and 71 lipids between these species, we acknowledge that there are many factors yet to be characterized in the *Lactobacillus* metabolic pool. Among the metabolites we identified, several hold potential to provide immune support and other benefits to the host in a probiotic relationship. We tested and verified the ability of *L. acidophilus* and *L. gasseri* to degrade oxalate even with availability of other carbon sources, providing supporting evidence for the need to further evaluate these *Lactobacillus* species as probiotic treatments for oxalate conditions. Further work is needed to fully define and characterize the *L. acidophilus* and *L. gasseri* metabolic profiles and validate their performance as oxalate-targeting probiotics.

## Supporting information

**S1 Fig. Principal component analysis scores plots for the metabolomic and lipidomic comparisons of *L. acidophilus* and *L. gasseri*.** PCA depicts clear separation and analytical distinction between the global metabolomes (A) and lipidomes (B) of *L. acidophilus* and *L. gasseri*. In the metabolomics analysis, 76.8% of the variance is explained in 2 PCs, mostly accounted for by PC1, and 89.0% explained in 5 PCs. In the lipidomics analysis 83.7% of the variance is explained in 2 PCs, mostly accounted for by PC1, and 94.0% explained in 5 PCs.
(TIF)

**S1 Table. All 97 identified metabolites (*m/z* & retention time match to analytical standard) with significance and magnitude of signal intensity difference between *L. acidophilus* and *L. gasseri*.** Significance: p≤1E-03.
(XLSX)

**S2 Table. All 71 identified lipids with significance and magnitude of signal intensity difference between *L. acidophilus* and *L. gasseri*.** Detection: 1 = Data-Dependent (top 5) MS/MS fragment *m/z* match, 2 = All-Ion-Fragmentation MS/MS fragment *m/z* match, 3 = Headgroup *m/z* match (class-ID). Significance: p≤1E-03.
(XLSX)

## Acknowledgments

This work was funded by the National Institutes of Health (Grant # 2R01DK088892-05A1). The authors would like to acknowledge Dr. Cory A. Leonard, University of Florida Department of Pathology, Immunology and Laboratory Medicine, for her assistance with sample generation, including media preparation and cell culture, harvest, and lysis. Also to be acknowledged are Dr. Jeremy P. Koelmel, Yale University Environmental Health Sciences Department, for his assistance with the lipidomics analysis, and Vanessa Y. Rubio, University of Florida Department of Chemistry, for her contribution to the graphical abstract. Data from this study is available at www.metabolomicsworkbench.org (data track ID 1809).

## Author Contributions

**Conceptualization:** Casey A. Chamberlain, Marguerite Hatch.

**Data curation:** Casey A. Chamberlain.

**Formal analysis:** Casey A. Chamberlain.

**Funding acquisition:** Marguerite Hatch, Timothy J. Garrett.

**Investigation:** Casey A. Chamberlain.

**Methodology:** Casey A. Chamberlain, Timothy J. Garrett.

**Project administration:** Timothy J. Garrett.

**Resources:** Marguerite Hatch, Timothy J. Garrett.

**Supervision:** Timothy J. Garrett.

**Writing – original draft:** Casey A. Chamberlain.

**Writing – review & editing:** Casey A. Chamberlain, Marguerite Hatch, Timothy J. Garrett.

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
