## [Decision Letter · Decision Letter 0]

18 Aug 2019

PONE-D-19-20190

Metabolomic Profiling of Oxalate-Degrading Probiotic Lactobacillus acidophilus and Lactobacillus gasseri

PLOS ONE

Dear Dr. Garrett,

Thank you for submitting your manuscript to PLOS ONE. After careful consideration, we feel that it has merit but does not fully meet PLOS ONE’s publication criteria as it currently stands. Therefore, we invite you to submit a revised version of the manuscript that addresses the points raised during the review process.

Both reviewers reported that the manuscript is interesting and well written. The authors are required to discuss the minor points raised by the reviewers.

We would appreciate receiving your revised manuscript by Oct 02 2019 11:59PM. To enhance the reproducibility of your results, we recommend that if applicable you deposit your laboratory protocols in protocols.io, where a protocol can be assigned its own identifier (DOI) such that it can be cited independently in the future. For instructions see: http://journals.plos.org/plosone/s/submission-guidelines#loc-laboratory-protocols

We look forward to receiving your revised manuscript.

Kind regards,

Andrea Motta

Academic Editor

PLOS ONE

Journal Requirements:

1. We note that you have stated that you will provide repository information for your data at acceptance. Should your manuscript be accepted for publication, we will hold it until you provide the relevant accession numbers or DOIs necessary to access your data. If you wish to make changes to your Data Availability statement, please describe these changes in your cover letter and we will update your Data Availability statement to reflect the information you provide.

2. Please include a copy of Table 2 which you refer to in your text on page 15

Reviewers' comments:

Reviewer's Responses to Questions

**Comments to the Author**

1. Is the manuscript technically sound, and do the data support the conclusions?

Reviewer #1: Yes

Reviewer #2: Yes

2. Has the statistical analysis been performed appropriately and rigorously? 

Reviewer #1: Yes

Reviewer #2: Yes

3. Have the authors made all data underlying the findings in their manuscript fully available?

Reviewer #1: Yes

Reviewer #2: Yes

4. Is the manuscript presented in an intelligible fashion and written in standard English?

Reviewer #1: Yes

Reviewer #2: Yes

5. Review Comments to the Author

Reviewer #1: In Methods

The authors should better clarify the storage times, were the bacterial lysates stored at -80 ° C for how long before the analysis?

Was the storage time the same for all the samples analyzed in this work?

The authors compare the metabolic profiles between the two bacterial strains grown in a medium supplemented with oxalate (20 mM), they believe that it is not necessary to carry out controls on cultures not supplemented with oxalate? Or have they already performed these experiments in the past?

The description of lipid extraction should be improved

Reviewer #2: In the manuscript by Chamberlain et al, the authors deal focus on the metabolomes and lipidome of the oxalate-degrading Lactobacillus acidophilus and Lactobacillus gasseri. These studies might be important to define the metabolic properties of Lactobacillus that allow these bacteria to interact with the host intestine and influence overall health

The studies are well designed and data presented clearly. There are some issues that will need to be addressed prior to publication, however

The authors should explain for how long the bacterial lysates were stored at -80 ° C? before analysis

Have the authors performed the metabolic profiles also on cultures not supplemented with oxalate?

6. PLOS authors have the option to publish the peer review history of their article (what does this mean?). If published, this will include your full peer review and any attached files.

Reviewer #1: No

Reviewer #2: No

---

## [Author Response · Author response to Decision Letter 0]

22 Aug 2019

Reviewer 1:

Comment:

Comment #1: The authors should better clarify the storage times, were the bacterial lysates stored at -80 ° C for how long before the analysis?

Thank you for your feedback. The lysates were only stored for a brief time (approximately 1 month) before extraction, a time which was consistent across all samples. In our experience, lysates of this nature suspended in KH2PO4-based lysis buffer are stable for metabolomics analyses for several years, so we are confident that this short storage period is inconsequential to the experiment. We have adjusted the text (highlighted below) as follows to make this more clear to the reader.

“Cell lysates were immediately frozen at -80 °C to ensure their stability and were briefly held frozen (approximately 1 month) until needed for extraction, all samples being stored for an equal period of time. In our experience, lysates of this nature suspended in KH2PO4-based lysis buffer are stable for metabolomics analyses for several years.”

Comment #2: Was the storage time the same for all the samples analyzed in this work?

Thank you for your feedback. Yes, the storage time was the sample for all samples analyzed in this work. Please see our response to Comment #1 in which we clarify this point in the adjusted text.

Comment #3: The authors compare the metabolic profiles between the two bacterial strains grown in a medium supplemented with oxalate (20 mM), they believe that it is not necessary to carry out controls on cultures not supplemented with oxalate? Or have they already performed these experiments in the past?

Thank you for your feedback. The purpose of this experiment was to profile and compare the metabolomes of L. acidophilus and L. gasseri, not necessarily to compare the metabolomes in an oxalate versus non-oxalate environment. Oxalate was included in the media simply because that is how we have historically cultured Lactobacillus, but it will grow in many different types of media so it is difficult to assign one media as its native environment. Therefore, we did not perform an analysis on Lactobacillus grown without oxalate since such a control was outside of the focus of this experiment. 

It would be interesting for future studies to compare how the metabolome of Lactobacillus changes in different types of media, but being that it grows in many different matrices, it would be difficult to establish what is the “native” media since there is no designated media that puts it into its native state. 

Comment #4: The description of lipid extraction should be improved.

Thank you for your feedback. We believe there may be some confusion that the lipid extraction was not explained given the placement of references to the Folch method and our previous work at the beginning of the Lipid Extraction section. Immediately following this first sentence, the lipid extraction is explained in significant detail. To make this more clear to the reader, we have inserted text (highlighted below) that indicates the beginning of the description of the lipid extraction.

“Lipids were extracted using a modified version of the Folch method (41) similarly to our previous work (38) using the following process which we describe in detail here. From each normalized cell lysate sample, 150 µL was transferred to a 12 mL glass vial. Extraction blanks were included for downstream data filtering using 150 µL lysis buffer and were treated identically to biological samples. To each sample, 20 μL of internal standard mix (Avanti Polar Lipids, Alabaster, AL, USA) was added followed by brief vortexing – LPC(17:0), PC(17:0/17:0), PG(14:0/14:0), PE(15:0/15:0), PS(14:0/14:0), TG(15:0/15:0/15:0), PI(8:0), SM(d18:1/17:0), CER(d18:1/17:0), DG(14:0/14:0), CL(15:0(3)-16:1), SO(d17:1), PAzePC, CER(Glycosyl(β) C12), BMP(14:0 (S,R)), LSM(d17:1), 5 μg/mL each in 2:1 chloroform:methanol. Next, 400 μL methanol was added to each sample followed by vortexing, then 800 μL chloroform was added. Samples were vortexed and incubated on ice for 20 min, with vortexing at 10 and 20 min, followed by addition of 200 μL water. Samples were briefly vortexed and incubated on ice for 10 min with vortexing at 5 and 10 min. Separation of the organic and aqueous layers was achieved by centrifugation at 3260×g, 4°C for 10 min. The organic (bottom) layer containing lipid content was transferred to a new 12 mL glass vial in two steps: first with removal of 800 μL of the original organic layer, followed by another removal of 400 μL after re-extracting the remaining aqueous layer with 400 μL 2:1 chloroform:methanol by incubating on ice for 10 min and centrifugation at 3260×g, 4°C for 10 min. Lipid extracts were dried under nitrogen at 30°C and reconstituted in 300 μL isopropanol. Samples were centrifuged at 3260×g, 4°C for 10 min to pellet any residual protein, and 250 μL supernatants were transferred to glass LC vials for UHPLC-HRMS analysis.”

Reviewer 2: 

Comment #1: The authors should explain for how long the bacterial lysates were stored at -80 ° C before analysis?

Thank you for your feedback. Please see our response to Reviewer 1 Comments #1 and #2, which addresses these same questions.

Comment #2: Have the authors performed the metabolic profiles also on cultures not supplemented with oxalate?

Thank you for your feedback. Please see our response to Reviewer 1 Comment #3, which addresses this same question.

---

## [Editor Report · Decision Letter 1]

29 Aug 2019

Metabolomic Profiling of Oxalate-Degrading Probiotic Lactobacillus acidophilus and Lactobacillus gasseri

PONE-D-19-20190R1

Dear Dr. Garrett,

We are pleased to inform you that your manuscript has been judged scientifically suitable for publication and will be formally accepted for publication once it complies with all outstanding technical requirements.

With kind regards,

Andrea Motta

Academic Editor

PLOS ONE

Additional Editor Comments (optional):

All comments have been addressed.
---

## [Editor Report · Acceptance letter]

13 Sep 2019

PONE-D-19-20190R1 

Metabolomic Profiling of Oxalate-Degrading Probiotic *Lactobacillus acidophilus* and *Lactobacillus gasseri*

Dear Dr. Garrett:

I am pleased to inform you that your manuscript has been deemed suitable for publication in PLOS ONE. Congratulations! Your manuscript is now with our production department. 

With kind regards,

on behalf of

Dr. Andrea Motta 

Academic Editor

PLOS ONE